# Genetic Distribution of Five Spinocerebellar Ataxia Microsatellite Loci in Mexican Native American Populations and Its Impact on Contemporary Mestizo Populations

**DOI:** 10.3390/genes13010157

**Published:** 2022-01-16

**Authors:** Rocío Gómez, Yessica S. Tapia-Guerrero, Bulmaro Cisneros, Lorena Orozco, César Cerecedo-Zapata, Elvia Mendoza-Caamal, Gerardo Leyva-Gómez, Norberto Leyva-García, Luis Velázquez-Pérez, Jonathan J. Magaña

**Affiliations:** 1Department of Toxicology, CINVESTAV-IPN, Mexico City 07360, Mexico; mrgomez@cinvestav.mx; 2Laboratory of Genomic Medicine, Department of Genetics, National Rehabilitation Institute-Luis Guillermo Ibarra Ibarra (INR-LGII), Mexico City 14389, Mexico; yessicasarai@gmail.com (Y.S.T.-G.); misael_207@hotmail.com (C.C.-Z.); nleyva@inr.gob.mx (N.L.-G.); 3Department of Genetics and Molecular Biology, CINVESTAV-IPN, Mexico City 07360, Mexico; bcisnero@cinvestav.mx; 4Laboratory of Immunogenomics and Metabolic Diseases, National Genomic Medicine Institute (INMEGEN), Mexico City 14610, Mexico; lorozco@inmegen.gob.mx (L.O.); emendoza@inmegen.gob.mx (E.M.-C.); 5Rehabilitation and Social Inclusion Center of Veracruz (CRIS-DIF), Xalapa, Veracruz 91097, Mexico; 6Department of Pharmacy, School of Chemistry, Universidad Nacional Autónoma de México (UNAM); Mexico City 04510, Mexico; leyva@quimica.unam.mx; 7Cuban Academy of Sciences, La Havana 10100, Cuba; velazq63@gmail.com; 8Department of Bioengineering, School of Engineering and Sciences, Tecnologico de Monterrey, Campus Ciudad de México (ITESM-CCM), Mexico City 14380, Mexico

**Keywords:** spinocerebellar ataxias, CAG repeats, Native American population, Mexican population, allelic distribution, large normal alleles

## Abstract

Spinocerebellar ataxias (SCAs) conform a heterogeneous group of neurodegenerative disorders with autosomal dominant inheritance. Five of the most frequent SCAs are caused by a CAG repeat expansion in the exons of specific genes. The SCAs incidence and the distribution of polymorphic CAG alleles vary among populations and ethnicities. Thus, characterization of the genetic architecture of ethnically diverse populations, which have undergone recent admixture and demographic events, could facilitate the identification of genetic risk factors. Owing to the great ethnic diversity of the Mexican population, this study aimed to analyze the allele frequencies of five SCA microsatellite loci (SCA1, SCA2, SCA3, SCA6, and SCA7) in eleven Mexican Native American (MNA) populations. Data from the literature were used to compare the allelic distribution of SCA loci with worldwide populations. The SCA loci allelic frequencies evidenced a certain genetic homogeneity in the MNA populations, except for Mayans, who exhibited distinctive genetic profiles. Neither pathological nor large normal alleles were found in MNA populations, except for the SCA2 pre-mutated allele in the Zapotec population. Collectively, our findings demonstrated the contribution of the MNA ancestry in shaping the genetic structure of contemporary Mexican Mestizo populations. Our results also suggest that Native American ancestry has no impact on the origin of SCAs in the Mexican population. Instead, the acquisition of pathological SCA alleles could be associated with European migration.

## 1. Introduction

The autosomal dominantly inherited polyglutamine spinocerebellar ataxias (polyQ-SCAs) constitute a group of neuromuscular disorders caused by a CAG trinucleotide repeat expansion in the coding region of different genes [1,2,3]. The abnormal expansion of CAG repeats results, in turn, in the synthesis of mutant toxic proteins bearing a -polyglutamine tract (polyQ) [4]. Epidemiologically, SCA subtypes present a relative low frequency worldwide (3 in 100,000 inhabitants), with SCA3 being the most recurrent subtype, followed by SCA2, SCA1, SCA6, and SCA7; all of which are classified as polyQ-SCA [5,6]. Nonetheless, the distribution and prevalence of polyQ-SCAs vary between ethnicities and geographic regions. For instance, SCA2 is highly prominent in Cuba, while SCA3 and SCA7 have remarkable frequencies in Portuguese speaking countries and Mexico, respectively [5,7,8,9]. Studies to determine the prevalence of polyQ-SCAs and the length of CAG alleles in founder populations might help to elucidate the origin and spread of the expanded alleles. Founder effects have been found to be involved in the appearance and spread of SCAs in the American continents and the Caribbean [7]. Understanding of the genetic structure through SCA loci of Native American populations, which have undergone a recent admixture and several demographic events, is critical in order to clarify their diversity and demographic history, and to possibly identify relevant genetic factors for biomedical traits [10]. Nonetheless, the impact of the ethnic component on the contemporary genetic composition of MNA populations remains to be analyzed in depth. Specifically, under-representation of ethnic minorities, such as Native American populations, Latinos, and Africans, might cause a bias predicting disease risk in a given population [11].

The Mexican population presents a complex genetic architecture based on the admixture between Native American inhabitants (>50%), European migrants that arrived from Spain and Southern Europe (>40%), and West Africans brought to Mexico through the trans-Atlantic slave trade (<2%) [12]; such ethnic fusion originated the group known as “Mestizos” 500 years ago. Nevertheless, the contribution of Mexican Native American (MNA) populations to the study of disease-associated alleles remains unexplored. The present study aimed to analyze the influence of MNA populations in shaping the allelic distribution of five different SCA microsatellite loci (SCA1, SCA2, SCA3, SCA6, and SCA7) in the contemporary Mexican Mestizo populations. Further analysis of SCA loci data enabled us to infer genetic relatedness between MNA populations; nonetheless, we found no connection between the MNA population and the origin of SCA pathologies in Mexico. Our results support the hypothesis that European migration could have introduced pathological SCA alleles to the Mexican population.

## 2. Materials and Methods

### 2.1. Subjects

The field work was conducted in nine different states of Mexico, namely Chihuahua (CHH), Guerrero (GRO), Michoacán (MIC), Morelos (MOR), Oaxaca (OAX), San Luis Potosí (SLP), Sonora (SON), Yucatán (YUC), and Veracruz (VER) (Figure 1). Blood samples were collected from 566 unrelated individuals who identified themselves as Mexican Native Americans (MNA) (Figure 1). These individuals belong to 11 different ethnic groups, representing five linguistic families. They have lived in the same community at least during the last three generations (parents and grandparents), further speaking the Native languages own their community. Prior studies have demonstrated that the individuals analyzed herein are carriers of 95 ± 5.7% of NA ancestry [13]. The Research Committee of the National Rehabilitation Institute approved the data and sample collection. All MNA individuals signed an informed consent letter before sample obtention. To respect their identity, the ethnic names used in the present study correspond to their self-identification, which were obtained from the National Institute of the Indigenous Peoples.

### 2.2. Molecular Genetic Analysis

Genomic DNA was isolated from peripheral blood leukocytes with the Gentra Puregene Blood kit (QIAGEN, Hilden, NRW, Düsseldorf, Germany), following the recommendations of the manufacturer. DNA samples (15 ng) were amplified using a Verity 96-Well Fast Thermal Cycler (Applied Biosystems, Carlsbad, CA, USA), with the reaction conditions and chimeric primers described previously [9,14]. The resulting amplicons were characterized by capillary electrophoresis on an ABI Prism 3730XL system (Applied Biosystems, Foster City, CA, USA), following the manufacturer’s instructions, as previously reported [9]. Those samples genotyped for SCA2 and SCA7 who exhibited a homozygous genotype, were further analyzed using a triplet repeat primed assay [15] in order to accurately determine the presence of large CAG (>130) alleles associated with SCA pathologies. 

### 2.3. Statistical Analyses

The allele frequency was evaluated by analysis of molecular variance (AMOVA) and population differentiation analyses (*R_st_* values) using Arlequin v.3.5 [16]. The *R_st_* values were assessed using 10,000 permutations, and then visualized in a multidimensional scaling plot (MDS) using SPSS v.20 [17]. *p*-values were adjusted for false discovery rates in the R software [18,19]. The Hardy–Weinberg expectation (HW) was calculated with Weir and Cockerham’s *F* statistics using Genètix v.4.05.2 with 10,000 permutations [20]. 

### 2.4. Comparative Data

Data from the literature were used to compare the allelic distribution of SCA loci between MNA and Mestizo populations with worldwide populations (Appendix A). The studies where the allelic frequency was not adequately reported (i.e., bar plots) were excluded from this analysis. Thirty-seven worldwide populations (*n* = 7532 individuals) were included in the database for further analyses. Such a database was built through random sampling so as to avoid bias.

## 3. Results

### 3.1. Allele Frequencies and Genetic Diversity of Five SCA Loci in Mexican Native American Populations

The CAG repeat allele frequencies of the SCA1, SCA2, SCA3, SCA6, and SCA7 loci were estimated in Mexican Native American (MNA) populations belonging to six linguistic families. The allelic distribution of these microsatellite loci was found to vary to a certain extent between MNA populations (Appendix A). Briefly, SCA1 and SCA3 loci showed the highest diversity of CAG repeat alleles, with 13 and 18 different alleles in MNA. In contrast, the SCA7 locus exhibited the lowest diversity, with only eight different alleles compared with the remaining SCA loci. Interestingly, all SCA alleles showed a lower diversity in MNA than the other worldwide populations. For SCA1, Mixes and Totonacs showed the highest number of CAG alleles (*k* = 10 alleles), while Mazahuas and *Rarámuris* (Tarahumaras) exhibited the lowest diversity (*k* = 6 and *k* = 7, respectively). Interestingly, the 31 CAG repeats allele was predominant in 7 out of 11 MNA populations analyzed. With respect to the SCA3 locus, Mayans (*k* = 13) and Nahuas-GRO (*k* = 11) exhibited the highest diversity of CAG repeat alleles, with the 23 CAG repeat alleles being the most frequent in all of the ethnicities genotyped. For SCA2, the most frequent allele was 22 CAG repeats in all of the MNA populations, while Mayans (*k* = 9) showed the highest number of CAG alleles. For SCA6, only 10 CAG alleles were identified in MNA, and *Rarámuris* exhibited the highest diversity (*k* = 8). Finally, for SCA7, the 10 CAG repeats allele was the predominant one in all of the 14 MNA analyzed. Of note, Nahuas-HGO were set apart from Nahuas-MOR, considering the SCA2 locus genetic data (*R_ST_* = 0.132; *p*-value = 0.006). Overall, the SCA loci allelic frequencies of the MNA populations were similar to that reported for the contemporary Mexican mestizo population [9]. These findings imply that the genetic structure of the contemporary Mexican population has been shaped, at least in part, by the Native American ancestry.

Regarding the genetic structure of the SCA loci in MNA populations, we found a remarkable deviation from HW in eight out of eleven populations genotyped (Appendix A) due to homozygous excess (*F_IS_* range: 0.123–0.253). 

### 3.2. Absence of SCA Pathological Alleles in Mexican Native Americans Populations

To provide insight into the origin of SCAs in the Mexican population, we search for the presence of SCA pathological alleles in MNA populations. None of the MNA populations genotyped harbored genetic vulnerability to SCAs, apart from a subject belonging to the Zapotec population, who carried a pre-mutated allele of 35 CAG repeats at the SCA2 locus (Appendix A). As SCA2 large normal alleles were rarely found in MNA populations (<3.5%), the case mentioned above appears to be an isolated event. Large-normal alleles of the SCA1 and SCA3 loci were found in almost all the MNA populations genotyped (frequencies of <5% and <1%, respectively), while SCA6 and SCA7 large normal alleles were not detected.

### 3.3. Genetic Interplay between Mexican Native American Populations

To unveil the genetic interconnection between MNA populations, the SCA loci genotyping data were analyzed using analysis of molecular variance (AMOVA). Significant differences among MNA populations, as well as between MNA populations, were found (global variation source) (Table 1). Then, diverse model simulations were run to evidence the relative impact of a given population on the global genetic diversity of MAN populations. Interestingly, the removal of the Mayan population caused significant differences between populations to disappear (percentage of variation 0.023, *p*-value = 0.946). Although the removal of Nahuas-HGO decreased the variation between populations (1.896%), such differences remained significant (*p*-value ≤ 0.0001). Next, MNA populations were divided according to their spoken language (Uto-Aztecan (Nahuas, Tarahumaras and Yaquis), Oto-Manguean (Zapotecos and Pames), Totonaca-Tepehua (Totonacas), Mixe- Zoquean (Mixes), and Macro-Mayan (Mayas)) and their geographic origin. Neither the linguistic nor the geographic criteria impacted the genetic diversity of MNA populations. data not shown). Next, to explore the diversity patterns of MNA populations in more detail, the SCA loci genetic data were analyzed using a population differentiation test based on *R_ST_* values (Appendix A). MNA populations were categorized by ethnicity, geography, and language family criteria. The ethnicity criterion plotted Mays apart from the remaining populations (Figure 2A), even when the Mestizo populations from the Central Valley of Mexico (CVM) and Tlaltetela (VER) were included (Figure 2B); these findings highlight the distinctive genetic background of Mayans. Mayans, Mazahuas, CVM Mestizos, Nahuas-Guerrero, and Tlaltetela Mestizos were scattered around a well-defined group positioned in the plot center, composed by Yaquis, Totonacs, Nahuas-Morelos, Pames, Popolucas, Zapotecs, Raramuris, and Mixes (Figure 2B). Utilization of the geographic criterion reinforced the genetic differences between Mayans and the remaining populations (Appendix A), with differences between Mayans and North populations being the most prominent (*p* ≈ 0.039). On the other hand, applying of the language affiliation criterion resulted in a cluster conformed by the Mixe-Zoquean, Oto-Manguean, Totonaca-Tepehuan, Uto-Aztecan, and Zapotec linguistic families, all of which were set up apart from the Macro-Mayan family (Appendix A). MDS plots show the analysis of each SCA microsatellite separately, which are depicted in Supporting Information (Appendix A).

### 3.4. Comparison of the SCA Loci Allelic Distribution between Mexican Native American and Worldwide Populations

In order to have a broader landscape about the SCA loci allelic distribution, our findings were contrasted with previous reports from worldwide populations, using a population differentiation analysis based on *R**ST* values (Appendix A). Concerning SCA1, two well-defined clusters were observed; the first included the Mestizos and all the MNA populations, except for Yoemem, which was placed in the second group encompassing European populations (Appendix A). The worldwide comparison of the SCA2 allelic distribution rendered a dense cluster in the center of the plot encompassing all populations, except for Yoruba (Appendix A). Of note, Zapotec, CVM-Mestizo, and Cuban populations exhibited a closed genetic distance (*R**ST* ≈ −0.008, *p*-value = 0.732) (Figure 3).

Analysis of the SCA3 showed a loose cluster in the center of the plot composed by both MNA and the worldwide population, with *Rarámuris* being plotted separately (Appendix A). It is worth to note the genetic proximity of Mestizos (CVM and Tlaltetela) with Finland and Portugal populations. The comparative analysis for the SCA6 locus failed to evidence any genetic proximity between MNA populations and the European ones (Appendix A). Nonetheless, this analysis confirmed the remarkable genetic distance between Nahuas from Guerrero and the remaining MNA populations (Appendix A). Finally, the comparison of the allelic distribution patterns of SCA7 sustained the distinctive genetic diversity of Mayans and evidenced the genetic proximity between Mestizos from Tlaltetela and Finland populations (Appendix A).

## 4. Discussion

In this study, we report our knowledge, for the first time, on the allelic distribution of five different SCA loci in Mexican Native American (MNA) populations. Further analysis of this data enabled us to ascertain the impact of Native American ancestry on the contemporary Mexican population structure, which could provide some clues for the geographic origin of SCA pathologies in Mexico. 

The SCA loci allelic frequencies found in MNA populations resembled that of the contemporary Mexican population [9], which indicates that Native American ancestry shaped, at least in part, the genetic structure of the latter population. However, a low diversity of SCA loci alleles was found in MNA populations compared with the contemporary Mestizo population [9], implying that the interethnic admixture among Spanish, Native Americans, and African lineages might be the mechanism underlying the high diversity of SCA alleles observed in the contemporary Mexican population.

Departure from the HW equilibrium due to heterozygote deficiency was not an unexpected result in MNA populations, considering the fact that this phenomenon is a signature of inbreeding populations. The MNA genetic architecture seems to be influenced by geographic barriers that keep them isolated [21]. MNA populations are also deeply rooted in their traditions, maintaining thus their native language and cultural practices, despite 500 years of miscegenation [22]. Both features have favored the contemporary Natives to maintain their ancestral connections and to settle down in geographic areas fairly close to those of their ancestors [21]. We speculate that these ancient practices might have promoted the inheritance of almost identical alleles, thereby increasing homozygosity and causing HW departure.

### 4.1. Searching for Large SCA Alleles in Mexican Native Americans

It is thought that SCAs prevalence in a given population is driven by the presence of intermediate and/or large normal alleles [23,24]. Consistent with this, we found a low frequency of large normal alleles and virtually no pathological alleles in MNA populations, apart from a pre-mutated SCA2 allele (CAG_35_) in the Zapotec group. Thus, the presence of SCA pathologies in the Mexican population, with a high prevalence of SCA2 and SCA7, could be the consequence of founding effects, as previously proposed [7]. Hypothetically, European settlers, who were predominantly Spaniards, could introduce the ancestral SCAs chromosomes to Mexico. The significant gene flow from Europe to Mexico has been demonstrated by the analysis of uniparental markers of the Y-chromosome non-recombining region (NRY), because miscegenation was sex-biased due to the arrival of only 6% of female migrants to the Americas during colonization [25]. NRY lineages carried by European colonizers (i.e., R1b, E1b, G, J, and I) have been found in Mexican Mestizos [26,27,28,29]. In fact, European ancestry has previously been linked to different SCA subtypes in the Americas (i.e., SCA2, SCA3, and SCA7) [5]. With respect to the presence of a SCA2 permutated allele in the Zapotec population, we hypothesized that this Native group might play a role in the origin of SCA2 in Mexico, by spreading pre-mutated alleles during the colonial period, because Oaxaca was a trade bridge connecting itself with other Mexican states (Campeche, Chiapas, and Veracruz), as well as with Guatemala and Cuba during the Viceroyalty of Mexico (1521–1821) [30]. Taking into consideration that colonization occurred first in Cuba (1511) and later in Mexico (1519) [31], and that SCA2 is highly prevalent in the former country, we speculate that Europeans who carried premutation alleles arrived firstly to Cuba and then to Oaxaca. Further haplotype analysis using uniparental markers as well as runs of homozygosity and identity-by-descent analyses are required to test this assumption. It is thought that European migration is implicated in the introduction of SCAs pathological alleles into the American continents and the Caribbean [7]. In fact, founder effects are the driving force behind the origin of SCAs worldwide. For instance, specific founder effects are responsible for the high prevalence of SCA3 observed in the Portuguese/Azorean island of Flores and Northeastern mainland [32].

### 4.2. Genetic Landscape of Mexican Native American Populations Using SCA Microsatellite Loci

Currently, diverse MNA populations coexist in the geographically wealthy territory of Mexico. The interplay among MNA populations through time could have tailored their present genetic structure. Thus, characterization of the genetic markers in these populations could broaden the picture about the origin and spread of the SCA expanded alleles. Furthermore, the study of genetic markers with biomedical relevance, such as the SCAs microsatellite loci, could help to trace the origin of these pathologies, as well as to identify the genetic factors relevant for biological processes of the MNA and Mestizo populations of today. Nonetheless, these studies should be reinforced by the analysis of ancestral markers.

The genetic distance of MNA populations depicted minor differences between them, exhibiting a subtle percentage of variation (3.29%), which is consistent with other studies [33]. The analysis of the SCA1 microsatellite loci separated the *Yoemem* population from six out of eleven MNA populations studied. This finding could be explained by the European ancestry, not necessarily Spanish, acquired by its connection with USA populations [34]. The heterozygous excess of SCA3 loci (*F_IS_* = −0.186) supports this hypothesis. Likewise, evaluation of SCA2 and SCA3 loci failed to provide a clear population differentiation of MNA populations. Noteworthy, analysis of the SCA2 locus showed a certain divergence between Nahua populations (*R**ST* = 0.132; *p*-value = 0.006), which could be the result of a differential genetic structure between them; Nahuas-GRO exhibited high genetic diversity (*k* = 4), while Nahuas-MOR showed an endogamy behavior (*k* = 2). It could be argued that the gene flow of Nahuas-GRO with European and African ancestries [26,35,36,37] and with neighbor ethnicities occurred to a greater extent than that of Nahuas MOR. On the other hand, the diagnosis of SCA6 microsatellite clustered all MNA populations, with the exception of Nahuas-GRO and Mayans. Such grouping suggests a continuous gene-flow between MNA groups and might also indicate the absence of cultural boundaries [22]; the latter is supported by the linguistic connection that took place between all indigenous groups (except Mayans). In fact, the genetic connection among Meso- and Arido-American populations has been extensively reported [12,13,22,38,39]. Furthermore, the genetic fusion between most of the MNA populations could reflect the hegemony of the Aztec Empire [22]. Although Nahuas-GRO and Mayans exhibited a distinctive diversity between one another (*p* = 0.028), the genetic distance between them was not considerable (*R**ST* = 0.037). Similar results have been previously reported using NRY markers [27]. It is worth mentioning that both populations have undergone a distinctive gene flow from Europe through the Dutch, English, and French pirate invasion that occurred at the Yucatan Peninsula between the XVI and XVIII centuries [27,40,41]. 

The Mayan population was also segregated from five out of eleven MNA populations genotyped upon analysis of the SCA7 microsatellite locus. The genetic distance between Mayans and Mixes, *Yoemem* and Zapotecans, exhibited only a marginal significance, which could be due to the geographic closeness between Yucatan and Oaxaca [42]. Both the whole-genome sequencing and haplotyping of NRY markers (i.e., R1b, 10%; G-M201, 6%; I-M258, 3%; and J-M305, 1%) support the European admixture in Mayans [13,40], while the European ancestry of Nahuas-GRO is low (4.2%) [35,36]. On the other hand, the *Yoemem* diaspora to the United States of America could explain, at least in part, the partial genetic segregation from Mayans [34]. On the other hand, the genetic distance between Mayans and Nahuas-GRO and *Rarámuris* were marked. A portion of these differences could be attributed to the linguistic families to which they belong. While *Rarámuris* belong to Uto-Aztecan, Mayans belong to the Macro-Mayan family [43]. With respect to Nahuas-GRO, their prominent African ancestry could lead them to acquire a distinctive genetic profile (~4%) [26,35,37].

The limitations of this study are that the SCA loci alleles are not the suitable genetic tool to reconstruct demographic and cultural histories of populations, because they are not neutral markers. Furthermore, differences in the sample size between populations could bias our conclusions.

## 5. Conclusions

In summary, our findings highlight the importance of analyzing MNA populations in order to better understand the influence of Native American ancestry on modelling the genetic structure of contemporary Mexican Mestizo populations. Furthermore, this study suggests that Native Americans had no influence on the origin of the most prevalent SCAs in Mexico, reinforcing the idea that the introduction of the pathological alleles into Mexico could be associated with European migrations.

## Figures and Tables

**Figure 1 genes-13-00157-f001:**
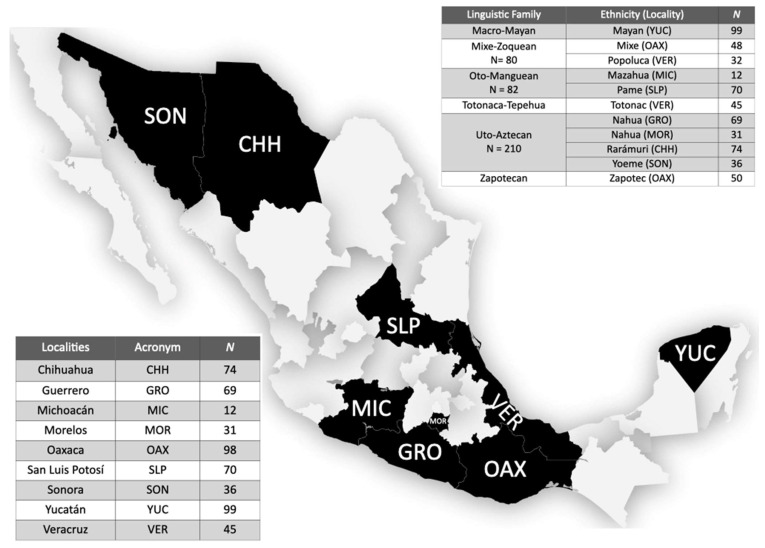
Map of Mexico showing the geographical origin of Mexican Native American populations included in the present study. The states where MNA populations live are shaded in black. *N*-number of subjects.

**Figure 2 genes-13-00157-f002:**
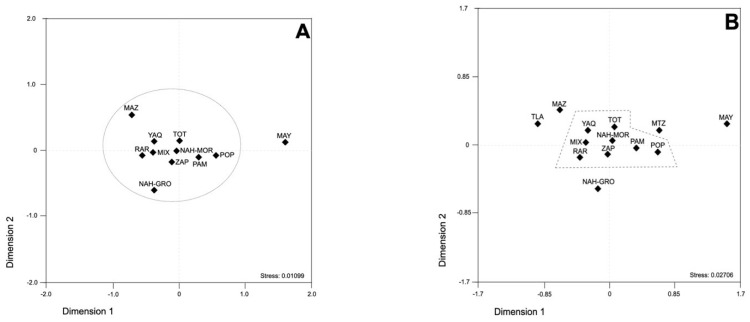
Multidimensional scale plot of *R**ST* values using five SCA microsatellite loci data, and the ethnicity criterion for contemporary Mexican Native Americans (**A**) and contemporary Mexican Native Americans and the Mexican Mestizos from the Central Valley of Mexico and Tlaltetela Veracruz (**B**). Mayans-MAY; Mazahuas-MAZ; Mixes-MIX; Nahuas-Guerrero-NAH-GRO; Nahuas-Morelos-NAH-MOR; Pames-PAM; Popolucas-POP; Rarámuris (Tarahumaras)-RAR; Yoemem (Yaquis)-YOE; Mexican Mestizos from the Central Valley of Mexico—CMV; Mexican Mestizos from Tlaltetela, Veracruz-TLA. Dotted lines enclose those populations that show genetic connections between them (no significant genetic distances); *p*-values were adjusted with the method of false discovery rates in R-software.

**Figure 3 genes-13-00157-f003:**
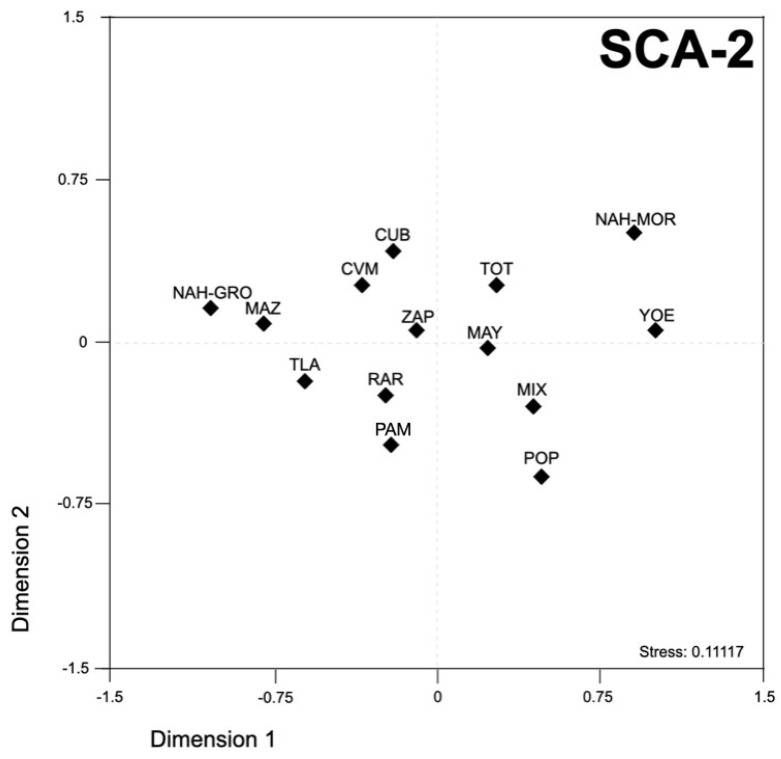
Multidimensional scale plot of *R**ST* values using the SCA2 locus data from Mexican Native American, Mestizos, and Cuban populations. Mayans-MAY; Mazahuas-MAZ; Mixes-MIX; Nahuas-Guerrero-NAH-GRO; Nahuas-Morelos-NAH-MOR; Pames-PAM; Popolucas-POP; Rarámuris (Tarahumaras)-RAR; Yoemem (Yaquis)-YOE; Mexican Mestizos from the Central Valley of Mexico-CMV; Mexican Mestizos from Tlaltetela, Veracruz-TLA; Cuba-CUB. *p*-values were adjusted with the method of false discovery rates in R-software.

**Table 1 genes-13-00157-t001:** Genotyping data from five different SCA loci of contemporary Mexican Native American populations subjected to the analysis of molecular variance.

Source of Variation	Variation (%)	*p*-Value
Global		
Among populations	3.29	0.004
Among individuals within population	52.43	≤0.0001
Within individuals	44.27	≤0.0001
Geography		
Among populations	4.02	0.0001
Among individuals within population	52.04	≤0.0001
Within individuals	43.94	≤0.0001
Language		
Among populations	3.99	≤0.0001
Among individuals within population	51.98	≤0.0001
Within individuals	44.03	≤0.0001

Geography: Centre (Nahuas-Morelos); North-East (Pames and *Rarámuris*); North-West (*Yoemem*); South (Mixes, Nahuas-Guerrero, Popolucas, Totonacs, and Zapotecs); South-East (Mayans); West (Mazahuas). Language: Macro-Mayan (Mayans); Mixe-Zoquean (Mixes and Popolucas); Oto-Manguean (Mazahuas and Pames); Totonaca-Tepehuan (Totonacs); Uto-Aztecan (Nahuas, *Rarámuris,* and *Yoemem*); Zapotecan (Zapotecs).

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
