# Peer review of "Genetic Distribution of Five Spinocerebellar Ataxia Microsatellite Loci in Mexican Native American Populations and Its Impact on Contemporary Mestizo Populations"

_genes, 2022, doi:10.3390/genes13010157_

Round 1
Reviewer 1 Report
In this study, the authors analyze allele frequencies of five SCA loci (SCA1, SCA2, SCA3, SCA6, and SCA7) in eleven Mexican Native Amerindian populations. These data are of main importance, namely on searching for large SCA alleles in Mexican Native Americans. The paper is well written and data analysis was properly performed. Also, the sample size is acceptable to this study.
Minor points:
In the Introduction, the authors state “… it appears that the ethnic background influences the incidence of polyQ-SCAs”; this should be rephrased since the ethnic background may not be influencing the frequency of SCAs, but may be only a reflection of the populations of origin and spread of expanded alleles.
In the Results (3.1. subsection), “CTG repeat” should be replaced by “CAG repeat”.
The Discussion would be enriched if the authors discussed their findings in the light of previous results on the origin and spread of SCAs (such as DOI: 10.1007/978-3-319-71779-1_12).
Major points:
The authors explain “remarkable deviation from the Hardy-Weinberg equilibrium (HWE) in eight out of eleven populations genotyped” (Supplementary Information, Table S7) due to homozygous excess. Why do the authors claim that HWE observed in MNA populations appears to be due to heterozygote deficiency rather than heterozygote excess? Is this assumption solely based on previous analysis of a large genomic dataset (as referenced by the authors; REF 21)? If so, the authors should bear in mind that in this large genomic dataset, sequencing technologies are responsible for false positive HetExc variants (which cannot be the problem in the manuscript under revision since the authors analysed CAG repeats by capillary electrophoresis.
In the topic 3.4, the authors compare the SCA loci allelic distribution between Mexican Native American and worldwide populations. The fact that the authors claim these comparisons to be an attempt to trace the geographic origin of SCA pathologies in the Mexican population seems as “beautification” of the potential reached by this strategy. Also, in the Discussion, the authors claim their results “provide clues for the geographic origin of SCA pathologies in Mexico”. Finally, in the Abstract, the hypothesis of founder events underlying the origin and spread of SCAs in Mexico seems also overrated.
Not having analysed any CAG expanded alleles, the authors should rewrite these statements in a more modest way.
Author Response
We thank the reviewers for their insightful comments and helpful criticisms. We have responded to all comments as detailed below and we now hope that you will find our revised Manuscript acceptable for publication in “Genes”.
REVIEWER 1
In this study, the authors analyze allele frequencies of five SCA loci (SCA1, SCA2, SCA3, SCA6, and SCA7) in eleven Mexican Native Amerindian populations. These data are of main importance, namely on searching for large SCA alleles in Mexican Native Americans. The paper is well written and data analysis was properly performed. Also, the sample size is acceptable to this study.
Minor points:
In the Introduction, the authors state “… it appears that the ethnic background influences the incidence of polyQ-SCAs”; this should be rephrased since the ethnic background may not be influencing the frequency of SCAs, but may be only a reflection of the populations of origin and spread of expanded alleles.
-We rephrased the mentioned sentence accordingly with the observation as follow: “Studies to determine the prevalence of polyQ-SCAs and the length of CAG alleles in founder populations might help to elucidate the origin and spread of the expanded alleles
In the Results (3.1. subsection), “CTG repeat” should be replaced by “CAG repeat”.
-The mentioned error was mended.
The Discussion would be enriched if the authors discussed their findings in the light of previous results on the origin and spread of SCAs (such as DOI: 10.1007/978-3-319-71779-1_12).
-We now included the suggested reference (see Reference 32) and added a brief discussion on it. We thank the reviewer for this opportune recommendation.
Major points:
The authors explain “remarkable deviation from the Hardy-Weinberg equilibrium (HWE) in eight out of eleven populations genotyped” (Supplementary Information, Table S7) due to homozygous excess. Why do the authors claim that HWE observed in MNA populations appears to be due to heterozygote deficiency rather than heterozygote excess? Is this assumption solely based on previous analysis of a large genomic dataset (as referenced by the authors; REF 21)? If so, the authors should bear in mind that in this large genomic dataset, sequencing technologies are responsible for false positive HetExc variants (which cannot be the problem in the manuscript under revision since the authors analysed CAG repeats by capillary electrophoresis.
- We thank the reviewer for this opportune observation. In concordance, we deleted this statement from Results, Section 3, and added instead a brief discussion with respect to the homozygous excess present in Mexican Native American populations (see Discussion, third paragraph, and references 21 and 22).
In the topic 3.4, the authors compare the SCA loci allelic distribution between Mexican Native American and worldwide populations. The fact that the authors claim these comparisons to be an attempt to trace the geographic origin of SCA pathologies in the Mexican population seems as “beautification” of the potential reached by this strategy. Also, in the Discussion, the authors claim their results “provide clues for the geographic origin of SCA pathologies in Mexico”. Finally, in the Abstract, the hypothesis of founder events underlying the origin and spread of SCAs in Mexico seems also overrated. Not having analysed any CAG expanded alleles, the authors should rewrite these statements in a more modest way.
-We totally agree with the reviewer's observation; in that our results did not demonstrate that the SCA loci allelic distribution of Mexican Native American populations is involved in the origin of SCAs in Mexico. In concordance, the referred statements were changed accordingly in the Results, 3.4 section, Discussion, and Conclusion sections.
Reviewer 2 Report
The manuscript titled “Genetic distribution of five Spinocerebellar Ataxia microsatel-2 lite loci in Mexican Native American populations and its im-3 pact on contemporary Mestizo populations” It may be reconsidered after major revision
First: Although there is originality in the work, the authors must review the text with an anti-plagiarism program to avoid excessive repetitions with their previous article and coincidences with other documents.
Second: The authors should make a good introduction and carefully review the conclusions based on the results obtained.
Third: Some results should not be expressed with such forcefulness with the number of individuals analyzed in such an area of land.
Author Response
We thank the reviewers for their insightful comments and helpful criticisms. We have responded to all comments as detailed below and we now hope that you will find our revised Manuscript acceptable for publication in “Genes”.
REVIEWER 2
The manuscript titled “Genetic distribution of five Spinocerebellar Ataxia microsatel-2 lite loci in Mexican Native American populations and its im-3 pact on contemporary Mestizo populations” It may be reconsidered after major revision
First: Although there is originality in the work, the authors must review the text with an anti-plagiarism program to avoid excessive repetitions with their previous article and coincidences with other documents.
- We followed this pertinent suggestion. The revised version was subjected to a CrossCheck analysis for homology with similar articles form the literature using iThenticate and Turnitin software. These analyses revealed only a 7% of homology with previous published studies. This low homology was mainly found in referred terminology and concepts.
Second: The authors should make a good introduction and carefully review the conclusions based on the results obtained.
- We thank the reviewer for this this opportune observation. We now built better the “Introduction” section and Conclusions were changed to reflect accurately our results.
Third: Some results should not be expressed with such forcefulness with the number of individuals analyzed in such an area of land.
- Following this meaningful observation, we carefully reviewed all the text and attenuated/toned down our statements and conclusions (please see Discussion and Conclusions).
Round 2
Reviewer 2 Report
The manuscript titled “Genetic distribution of five Spinocerebellar Ataxia microsatel-2 lite loci in Mexican Native American populations and its impact on contemporary Mestizo populations” It may be reconsidered after major revision
Please modify the last part of the introduction referring to the conclusion in line with the final conclusion of the article:
“Collectively our findings demonstrated contribution of the Amerindian ancestry in shaping the genetic structure of contemporary Mexican Mestizo populations, and support the hypothesis that founder events, occurring after the Spaniard colonization of Mexico, underlie the origin and spread of SCAs in Mexico.”
“… could be associated with European migrations”.
In case of disagreement, justify thoroughly and objectively, the events to which they refer. If you decide to do this, you must Increase the number of individuals analyzed to justify your conclusions.
Author Response
REBUTTAL LETTER
REVIEWER 1
The manuscript titled “Genetic distribution of five Spinocerebellar Ataxia microsatellite loci in Mexican Native American populations and its impact on contemporary Mestizo populations” It may be reconsidered after major revision
Please modify the last part of the introduction referring to the conclusion in line with the final conclusion of the article: “Collectively our findings demonstrated contribution of the Amerindian ancestry in shaping the genetic structure of contemporary Mexican Mestizo populations, and support the hypothesis that founder events, occurring after the Spaniard colonization of Mexico, underlie the origin and spread of SCAs in Mexico.” “… could be associated with European migrations”.
In case of disagreement, justify thoroughly and objectively, the events to which they refer. If you decide to do this, you must Increase the number of individuals analyzed to justify your conclusions.
-We totally agree with the reviewer's observation in that our data did not demonstrated any connection between the genetic structure of Native Americans and the origin of SCAs in Mexico. In concordance, the last sentence of “Introduction” is in indeed in line with the Conclusion (please see the two sentences below.
Introduction. “Further analysis of SCA loci data enabled us to infer genetic relatedness between MNA populations; nonetheless, we found no connection between MNA population and the origin of SCA pathologies in Mexico. Our results support the hypothesis that European migrations could have introduced pathological SCA alleles to the Mexican population.
Conclusion. Furthermore, this study suggests that Native Americans had no influence on the origin of the most prevalent SCAs in Mexico, reinforcing the idea that introduction of the pathological alleles into Mexico could be associated with European migrations.
We found that the sentence referred by the reviewer, which required to be modified, is located in the last paragraph of “Abstract”.
Thus, following the pertinent observation of the reviewer, we mended this sentence to be in line with both “Introduction” and “Conclusion”, as follows:
Our results also suggest that the Native American ancestry has no impact on the origin of SCAs in the Mexican population. Instead, the acquisition of pathological SCA alleles could be associated with European migrations”.
We hope this sentence make clear our conclusions.